# Small means immoral? The impact of spatial size metaphor on moral judgment

**Weirui Xiong**⊙*, **Jiaxin Wang, Jiayi Li**

School of Educational Science, Chongqing Normal University, Chongqing, China

* xinyun0501@163.com

## Abstract

This study aims to explore the unconscious relationship between moral concepts and the spatial dimension of size, as well as to examine whether the unknown size of a room influences participants' moral cognitive judgments within the framework of embodied cognition. Study 1 and Study 2 investigate participants' unconscious biases. Specifically, participants exhibited faster response times when judging moral concepts presented in large fonts and sizes and immoral concepts presented in small fonts and sizes, compared to when moral concepts were presented in small fonts and sizes and immoral concepts in large fonts and sizes. Study 3 employed a moral dilemma task, revealing that participants placed in a large room evaluated characters in a story more morally under the embodiment effect than those in a small room. Collectively, these three studies demonstrate that the unconscious psychological relationship between moral concepts and the spatial dimension of size influences individuals' abstract moral judgments under embodied cognition.

## Introduction

In the process of social development, morality serves as a core element of human behavior and values. Its significance lies in regulating individual conduct and maintaining social order. But how do we comprehend the abstract concept of morality in our daily lives? In what ways do we associate moral concepts with tangible aspects of life?

Within the framework of embodied cognition, Lakoff and Johnson [1] proposed the Conceptual Metaphor Theory (CMT), which suggests that the conceptual systems underlying human thought, cognition, and behavior are structured by metaphors. This theory posits that understanding occurs through a process of mapping one domain onto another. Specifically, two key domains—the source concept and the target concept—are mapped onto each other to facilitate comprehension. A source concept originates from a direct experience in a person's daily life, whereas a target concept is an abstract and vague idea, entity, or emotion [1]. According to CMT,

**Data availability statement:** All relevant data are within the paper and its Supporting information files.

**Funding:** The author(s) received no specific funding for this work.

**Competing interests:** The authors have declared that no competing interests exist.

understanding target concepts is facilitated by drawing on well-known and clearly represented prior experiences (source concepts).

In the Chinese linguistic landscape, the use of size-related words is closely associated with emotional valence. The adjectives "big" *(da)* and "small" *(xiao)* frequently denote individual moral attributes. For instance, in Chinese, phrases such as *da du* (generous), *xiao qi* (petty), *da xiong huai* (large-heartedness), and *xiao xin yan* (pettiness) are commonly used to describe an individual's moral disposition. Similarly, in English, certain words such as large, wide, and broad are used to describe moral behavior, while words such as small, narrow, and thin are often associated with immoral behavior. People comprehend abstract concepts not only through concrete notions of size but also by recognizing the correlation between size and emotional valence.

Meier, Robinson, and Caven [2] investigated the correlation between the conceptualization of size and affective valence. Their study found that participants could understand abstract concepts through concrete representations of size. Specifically, the results showed that larger stimuli (as opposed to smaller stimuli) were associated with more positive moral evaluations. For example, in studies 1 and 2, when positive words were presented, participants judged words in larger fonts with faster reaction times, quicker evaluations, and higher accuracy compared to words in smaller fonts. The opposite trend was observed for negative words. Building on these findings, Study 3 demonstrated that the larger the font, the more positively the word was evaluated, indicating that the size effect influences lexical valence judgments. The larger font of the words conveyed a more positive meaning, illustrating the metaphorical association between 'big and positive' and 'small and negative". Martijn, Henk, and Ruud [3] examined the desire to achieve the goal and found that participants would increase the perceived size of an object, which would help achieve the goal. For instance, the researchers hypothesized that when a participant was deprived of the freedom to drink water, they would perceive a cup as larger. When the behavioral goal concept was paired with positive emotion, the motivation to engage in the action goal increased, resulting in the target tool used (e.g., shovel) being perceived to have a bigger size.

With the development of experiential philosophy and phenomenology, Merleau-Ponty [4] emphasized that the importance of the body and senses arises within organizational behavior. Furthermore, conceptual representation, as a fundamental and primary cognitive ability of humans, is inherently linked to the individual's perceptual-motor system and cannot exist independently of it [5]. Embodied cognition refers to the necessity of using somatic experience to comprehend complex problems and abstract concepts. Moral judgment is fundamentally a process of abstract value-based cognitive assessment. Research on conceptual metaphor has demonstrated that an individual's somatic perceptions influence cognitive judgments through metaphorical mediation. Witt [6] found that the size of paddles used to block balls can influence how quickly users perceive the speed of the balls. Lakens, Semin, and Garrido [7] identified a metaphorical correlation between individuals' conceptualization of time and horizontal space, where the "left" is associated

with the past and the "right" with the future. This finding suggests that embodied bodily perceptions influence abstract value judgments. Regarding the experimental procedure, participants were required to provide an optimal rating for each word. Within the framework of the purification metaphor, Ding and Wang [8] observed that participants primed with self-dirtiness exhibited heightened reactivity to negative moral emotion words compared to those primed with self-cleansing. However, no significant differences were found in participants' judgment responses to non-moral emotional words following self-cleansing or self-dirtiness priming. This suggests that the concept of self-dirtiness, influenced by bodily experiences, facilitates the processing of negative moral emotion words. Additionally, in the experimental procedure, participants were not given a time limit to respond to the words, suggesting that they had ample time to process them thoroughly. Lu, Guo, and Jiang [9] examined the existence of a consistent psychological phenomenon linking moral concepts with the dimension of size. The first two experiments in their study revealed that participants responded more quickly when moral terms were presented in a large font. Conversely, categorical assessments were inhibited when moral words appeared in a small font. Additionally, participants exhibited faster judgment responses to immoral words displayed in a small font. In their research procedure, participants were given 2500 ms to respond to each word. Most studies are conducted at the conscious level. Therefore, the existence of an unconscious mental association with moral conceptual size-space metaphors has yet to be empirically substantiated, nor has their metaphorical consistency been rigorously examined within the context of complex, embodied moral judgment tasks. Consequently, Study 1 examined the psychological reality of metaphors linking moral concepts to large and small spaces at an unconscious level using a variant of the Stroop paradigm. In an unconscious context, participants named matching patches more quickly than non-matching ones, demonstrating the typical Stroop effect [10]. Study 2 used the variation of the Implicit Association Test (IAT) paradigm to explore further at an unconscious level whether the processing of moral concepts links to the participant's perception of big and small spaces. The purpose of IAT is to establish the intrinsic connection between the target concept and the source concept so as to evaluate people's implicit attitude towards a specific object Participants are required to classify pairs of stimuli into mutually exclusive categories. Without prior evaluations of the target concept or source concept, performance on this task is expected to be equal for both conditions. However, if participants have some pre-existing associations of the target concept with the source concept, their pre-existing associations can interfere with the classification task [11]. Both experiments controlled reaction time to prompt quick responses from participants, allowing for the inference of unconscious reactions.

In metaphorical studies of embodied moral concepts, prior research has primarily focused on weight metaphors [12–16], color metaphors [17–19], and temperature metaphors [20–22], among others. However, to date, few studies have systematically examined the role of size-space metaphors in the context of embodied moral concepts. Embodied cognition theory holds that cognitive activities are not only inseparable from the brain but are also closely linked to the interaction between the body and the environment. The brain is embedded in the body, and the body is embedded in the environment, forming an integrated cognitive system [23]. Research has shown that individuals represent and process conceptual knowledge through simulations of sensorimotor experiences [24]. For instance, when processing the concept of "apple," the brain activates multimodal sensory information, including visual (red color), tactile (smooth texture), and gustatory (sweet taste) inputs, to construct an understanding of the concept. Consequently, Study 3 delves into the exploration of how size-space perceptions link to an individual's moral judgments by unconsciously activating the participants' perceptual information.

In conclusion, the linguistic manifestation of moral concepts through size-space metaphors is well-documented. Scholars, including Lu et al. [9], have substantiated the psychological validity of size metaphors at the conscious level. Consequently, the present study aims to explore the unconscious realm to investigate the psychological foundations of the metaphorical representation of large and small spaces. Furthermore, it seeks to determine whether the uncertainty of room size influences participants' moral cognitive judgments within the framework of embodied cognition. These studies aim to expand knowledge on how humans make moral judgments and how these judgments are

embodied. The anticipated contributions lie in bridging gaps between unconscious cognitive processes and embodied metaphor theory, potentially informing interdisciplinary discussions in moral psychology, linguistics, and environmental design.

## Study 1: The psychological reality of spatial metaphors of size for moral concepts through Stroop

### Participants

According to calculation using G*power 3.1 [25], for the two-way repeated measures analysis of variance applicable to this study, with a significance level of α = 0.05, a large effect size (*f* = 0.4), and a predicted statistical power (1-β) of 0.8, the required total number of research subjects was 22. We recruited 43 university students, including 25 women. All participants provided informed consent before the experiment. All participants were right-handed, had normal or corrected-to-normal visual acuity (>1.0), met the experimental criteria, and did not have dyslexia. All studies were approved by the Ethics Committee of Chongqing Normal University. All informed consent forms were signed before participants completed the questionnaire for evaluation and began the studies. All experimental procedures adhered to the ethical guidelines outlined in the Declaration of Helsinki.

### Materials and methods

The experimental materials consisted of moral and immoral words selected from the *Dictionary of Modern Chinese Frequency*. A separate group of 34 participants who did not take part in the formal experiment assessed the familiarity and moral valence of these words using a 7-point Likert scale (1 = very immoral to 7 = very moral).

The morality scores of the moral words were significantly higher than the midpoint of the scale (4) (*M* = 6.23, *SD* = 0.56), while the morality scores of the immoral words were significantly lower than the midpoint (*M* = 1.78, *SD* = 0.45), with a statistically significant difference, $t(33) = 36.05$, $p < 0.001$. The mean familiarity scores for both moral and immoral words were above 5, and the difference in familiarity scores between moral words (*M* = 6.03, *SD* = 0.81) and immoral words (*M* = 5.70, *SD* = 0.75) was not statistically significant, $t(33) = 1.76$, $p = 0.082$. Based on these evaluations, 20 moral and 20 immoral words were ultimately selected as experimental materials.

### Research design

The experiment employed the spatial Stroop paradigm within a two-factor, within-subjects design, structured as a 2 (lexical type: moral words, immoral words) × 2 (font size: large font, small font) factorial arrangement.

### Procedure

The experiment was programmed using E-Prime 2.0, consisting of both practice and formal experimental sessions.

To initiate the task, a red "+" attention cue appeared at the center of the screen for 500 milliseconds. Subsequently, 40 words were presented randomly, one at a time, in the center of the screen. Each word appeared twice, varying in font size and sequential position. Participants categorized words as moral by pressing the "F" key and as immoral by pressing the "J" key. The assignment of response keys for moral and immoral classifications was counterbalanced across participants. If a response was not made within 1500 milliseconds, the word automatically disappeared, followed by the reappearance of the red "+" cue for another 500 milliseconds to signal the next trial. Prior to commencing the formal experiment, participants underwent an 8-trial practice phase to acclimatize themselves to the experimental procedures and operational protocols. They indicated their preparedness to initiate the formal experiment by pressing the "P" key. Participants unfamiliar with the task could return to the practice phase at any point by pressing the "Q" key to review the process before beginning the formal experiment. The experimental system automatically recorded all relevant data.

## Data analysis

**Data screening and processing.** After the experiment was completed, data were processed following the standard requirements of the Stroop paradigm [10] using SPSS 20.0. Participants whose accuracy rate fell below 80% were excluded from the dataset. Additionally, response times that deviated by more than 2.5 standard deviations from the mean were removed. The exclusion process ensured that no more than 3% of the total data were discarded, preventing any significant loss of data. The remaining valid dataset, consisting of 35 participants, was analyzed using repeated measures ANOVA.

**Descriptive statistical analysis.** Descriptive statistics of participants' response times and accuracy rates for different lexical judgment conditions are presented in Table 1.

As shown in Table 1, when the font size was consistent, the judgment reaction time for moral materials (567±53.2 ms) was significantly quicker than that for immoral materials (620±85.6 ms). Conversely, when the type of vocabulary materials was consistent, varying font sizes influenced participants' judgments, with moral words in a large font being processed significantly faster (567±53.2 ms) than those in a small font (587±67.1 ms). Additionally, the reaction time for immoral words in a small font (599±66.9 ms) was significantly faster than that for immoral words in a large font (620±85.6 ms).

When the font size was consistent, moral words were judged correctly more often (0.98±0.04) than immoral words (0.93±0.03). When the type of vocabulary materials was the same, moral words in a large font (0.98±0.04) were judged correctly more often than those in a small font (0.96±0.04). Similarly, immoral words in a small font (0.96±0.03) were judged correctly more often than those in a large font (0.93±0.03).

**Analysis of variance for repeated measurements of reaction time.** The dependent variables of the experiment were the response time and accuracy of participants' judgments of the lexical material. ANOVA results are presented in Table 2. The main effect of lexical word type was significant: in the task involving different lexical types of material, participants judged moral words (577 ms) faster than immoral words (609 ms), $F(1,34) = 24.622$, $p=0.000<0.01$, $\eta^2_p=0.420$.

The interaction between vocabulary type and font size was significant, $F(1,34) = 19.858$, $p=0.000<0.01$, $\eta^2_p=0.369$. For moral words, participants' categorical judgment of words presented in a large font was significantly faster than for those presented in a small font, $F(1,34) = 9.708$, $p=0.004<0.01$, $\eta^2_p=0.222$. For immoral words, participants' categorical judgment of words presented in a small font was significantly faster than for those presented in a large font, $F(1,34) = 10.179$, $p=0.003<0.01$, $\eta^2_p=0.230$.

**Table 1. Different vocabulary type judgments with words.**

|  | Moral Words + Big Font Size | Moral Words + Small Font Size | Immoral Words + Big Font Size | Immoral Words + Small Font Size |
|---|---|---|---|---|
| Response Time (ms) | 567±53.2 | 587±67.1 | 620±85.6 | 599±66.9 |
| Accuracy Rate | 0.98±0.04 | 0.96±0.0 4 | 0.93±0.03 | 0.96±0.03 |

**Table 2. Response times in different task conditions.**

| Source of variation | F | p | η2p |
|---|---|---|---|
| Lexical Category | 24.622 | 0.000 | 0.420 |
| Font Size | 0.027 | 0.870 | 0.001 |
| Vocabulary Type * Font Size | 19.858 | 0.000 | 0.369 |

**Table 3. Accuracy in different task conditions.**

| Source of variation | F | p | η2p |
|---|---|---|---|
| Lexical category | 12.320 | 0.001 | 0.266 |
| Font size | 1.796 | 0.189 | 0.050 |
| Vocabulary type * Font size | 12.803 | 0.001 | 0.274 |

**ANOVA for repeated measures of accuracy.** A repeated measures ANOVA was conducted on participants' accuracy; the results are presented in Table 3. The main effect of vocabulary type was significant: participants were more accurate when judging moral words (97%) than when judging immoral words (94%), $F(1,34)=12.320$, $p=0.001<0.01$, $\eta^2_p=0.266$.

The interaction between vocabulary type and font size was significant, $F(1,34) = 12.803$, $p=0.001<0.01$, $\eta^2_p=0.274$. The results indicate that when the experimental task involved immoral words, participants made correct judgments more frequently with a small font than with a large font, $F(1, 34) = 13.222$, $p=0.001<0.01$, $\eta^2_p=0.280$.

In summary, there is a psychological association of "morality with big and immorality with small." Study 2, building on these findings, will further investigate the psychological validity of spatial metaphors of size for moral concepts using the IAT.

## Study 2: The psychological reality of spatial metaphors of size for moral concepts through IAT

### Participants

According to calculation using G*power 3.1 [25], for the Paired samples t-tests applicable to this study, with a significance level of α = 0.05, a large effect size ($d=0.8$), and a predicted statistical power (1-β) of 0.8, the required total number of research subjects was 15. A total of 48 current university students were selected as experimental participants, including 28 women participants, meeting the same experimental criteria as above. All informed consent forms were signed before the experiment. This study was approved by the Ethics Committee of Chongqing Normal University. All informed consent forms were signed before participants began the study. All experimental procedures adhered to the ethical guidelines outlined in the Declaration of Helsinki.

### Materials and methods

The materials comprised two types of concepts: target concepts and source concepts. The target concepts included 10 moral and 10 immoral words, while the source concepts consisted of two squares of different sizes (8 cm x 8 cm and 4 cm x 4 cm). All materials were presented on a computer screen. Thirty-two students who were not formal participants in this experiment were selected to rate the morality and familiarity of these target and source concepts. The results indicated that the moral valence scores for the moral words in the target conceptual material were significantly greater than 4 on a 7-point scale, where 1 indicated "very immoral" and 7 indicated "very moral" ($M=5.80$, $SD=0.68$). In contrast, the scores for the immoral words were significantly lower than 4 ($M=2.26$, $SD=0.85$). The difference in moral valence between the two types of vocabulary was statistically significant, $t(31) = 18.27$, $p<0.001$. In terms of familiarity, the mean score was greater than 5, and there was no significant difference in the scores between moral and immoral words, $t(31) = 0.94$, $p=0.348$. Regarding the source conceptual material, the difference in size scores between the large squares ($M=5.28$, $SD=1.25$) and the small squares ($M=3.15$, $SD=1.43$) was significant, $t(31) = 6.30$, $p<0.001$. However, there was no significant difference in familiarity scores between the two sizes, $t(31) = 0.86$, $p=0.39$.

### Research design

A variant of the IAT paradigm was employed to categorize the target concepts. The participants' tasks included compatibility and incompatibility tasks. In the compatibility tasks, participants were instructed to classify moral words with large

**Table 4. Specific procedure for IAT.**

| Mandates | Keyboard Q | Keyboard P |
|---|---|---|
| Vocabulary Categorization | Moral word | Immoral word |
| Source Concepts | Big square | Small square |
| Compatibility Joint Tasks (exercise) | Moral word/big square | Immoral word/ small square |
| Compatibility Joint Missions (test) | | |
| Contrary Target Concept Identification | Immoral word | Moral word |
| Incompatibility Joint Tasks (exercise) | Immoral word/big square | Moral word/small square |
| Incompatibility Joint Missions (test) | Immoral word/big square | Moral word/small square |

**Table 5. IAT procedure step.**

| Segment | Number of tests | Functionality | Reaction | |
|---|---|---|---|---|
| | | | Left click | Right-click |
| 1 | 20 | Exercise | Moral words | Immoral words |
| 2 | 20 | Exercise | Big square | Small square |
| 3 | 40 | Exercise | Moral words + big squares (left and right) | Immoral words + small squares (left and right) |
| 4 | 40 | Test | Moral words + big squares (left and right) | Immoral words + small squares (left and right) |
| 5 | 20 | Exercise | Immoral words | moral words |
| 6 | 40 | Exercise | Immoral words + big squares (left and right) | Moral words + small squares (left and right) |
| 7 | 40 | Test | Immoral words + big squares (left and right) | Moral words + small squares (left and right) |

squares and immoral words with small squares. Conversely, in the incompatibility tasks, participants were required to classify moral words with small squares and immoral words with large squares. The specific seven-step procedure for this experiment is outlined in Table 4.

## Procedure

The experimental procedure was programmed using E-Prime 2.0 (refer to Table 5). After the formal experiment was initiated, a gaze cue consisting of a red "+" appeared at the center of the screen for 500 ms. This was followed by the presentation of the target concept, at which point participants were instructed to execute keystroke responses for the compatibility and incompatibility tasks in accordance with the experimental instructions during the trial period (Button presses had to be completed within 1500 ms or the interface would advance to the next screen.). Subsequently, participants were given a 1-minute rest after completing the three task processes before proceeding to the next phase of the experiment, which included the practice task. The experiment consisted of a practice task and a test task, with the study focusing exclusively on recording and analyzing the test data for the combined task.

## Data analysis

**Data screening and processing.** The data were processed following Greenwald's methodology for IAT experimental data [26] using SPSS 20.0. Initially, participants with an accuracy rate below 80% were excluded, as were those whose response times exceeded three standard deviations above 300 ms. Next, data entries with incorrect responses were removed. Notably, at least 3% of the total experimental data were retained for analysis. Ultimately, the valid dataset consisted of test data from the formal experimental tasks. Following this screening process, 39 valid data points that met the experiment's criteria remained.

**Table 6. Response time (ms) and accuracy on the IAT.**

| Task | M±SD | t | p |
|---|---|---|---|
| Response time | | | |
| Compatibility task | 534.32±93.16 | −7.786 | <0.001 |
| Incompatibility task | 611.20±72.12 | | |
| Accuracy rate | | | |
| Compatibility task | 0.96±0.34 | 3.88 | <0.001 |
| Incompatibility task | 0.93±0.43 | | |

**Statistical analysis of response time and accuracy rate.** Paired samples t-tests were conducted on the fourth step of the compatibility joint task (i.e., associating moral words with large squares and immoral words with small squares) and the seventh step of the incompatibility joint task (i.e., associating moral words with small squares and immoral words with large squares) within the formal test joint task data. This was done to compare participants' mean response times and accuracy rates between the joint tasks. The results are presented in Table 6.

There was a significant difference in participants' reaction times and accuracy rates between the compatibility joint task and the incompatibility joint task. The difference in reaction time was statistically significant, $t = -7.786$, $p < 0.001$, indicating faster responses in the compatibility task (534.32 ms) compared to the incompatibility task (611.20 ms). Similarly, the accuracy rate was significantly higher in the compatibility task (96%) than in the incompatibility task (93%), $t = 3.88$, $p < 0.001$.

**IAT statistical analysis.** This experiment employed the implicit paradigm to analyze the data and calculate the indicator D-value, utilizing the updated methodology for handling D-values proposed by Greenwald [27]. This approach not only considers the difference in response times between the formal test joint task and the non-joint task but also incorporates data from the practice joint task into the calculation of participants' response time differences. The data from the practice joint task (steps 3 and 6) and the formal test joint task (steps 4 and 7) were analyzed. First, the mean reaction times for the practice tasks and the formal test tasks were determined (M3, M6, M4, M7). Subsequently, the differences in reaction times between the compatibility and incompatibility tasks during the practice joint phase and the formal test joint phase were calculated (M6-M3; M7-M4). The joint standard deviation for the reaction times of the practice task step was denoted as S1, and that for the formal test task step was denoted as S2. Consequently, D1=(M6-M3)/S1 and D2=(M7-M4)/S2 were calculated, respectively, with the final D-value being determined as D=(D1+D2)/2.

A calculated D-value greater than zero signifies a stronger implicit association effect for the experiment, whereas a value of zero indicates a neutral implicit association and a value less than zero suggests the absence of an implicit connection. In this study, D was 0.35, indicating a significantly strong IAT effect. The relationship between these associations has been validated at the conscious level in previous studies and at the unconscious level in the two preceding studies within this paper. Consequently, Study 3 will further explore the unconscious influence of bodily experiences on moral judgments.

## Study 3: Effects of different sizes of spatial perception on moral judgment

### Participants

According to calculation using G*power 3.1 [25], for the independent samples t-test applicable to this study, with a significance level of α=0.05, a large effect size (d=0.8), and a predicted statistical power (1-β) of 0.8, the required total number of research subjects was 52. A total of 60 university students were selected for Study 3, with 30 assigned to the large room group and 30 to the small room group, all meeting the same experimental criteria as previously described. All informed consent forms were signed before the experiment. This study was approved by the Ethics Committee of Chongqing Normal University. All informed consent forms were signed before participants began the study. All experimental procedures adhered to the ethical guidelines outlined in the Declaration of Helsinki.

## Materials and methods

Ten stories were selected as the material for this experiment, following the example of prior studies that used stories as experimental materials [28–30]. 40 students who did not participate in the formal experiment were selected to rate the moral dilemma stories on a 7-point scale of difficulty, with 1 indicating "very easy to judge whether the character's behavior was moral or immoral" and 7 indicating "very difficult to judge." A sample story is presented below:

> The ship you are traveling on has hit a reef and is sinking fast, so you have to abandon the ship to survive. However, the life rafts available for escape have limited capacity, and your life raft is already sinking because it is overloaded with people. Something must be done, or everyone on board will drown. The captain decides to throw the crew, who are already seriously injured, overboard. Doing so will save the rest of the people. If nothing is done, the life raft will surely sink due to the excess weight, and everyone will die.

> The statistical results showed that the difficulty score of the moral dilemma story materials was $M = 4.29$, $SD = 1.02$, $t(39) = 26.50$, $p < 0.01$, suggesting that the selected moral dilemma story materials were representative. This implies that participants found it more challenging to make moral judgments about the characters in the stories.

## Research design

The experiment employed a one-way between-subjects design, with the independent variable consisting of two spaces of differing sizes: a large room and a small room. The dependent variable was the participants' ratings of the behavior of the characters in the moral dilemma stories. Participants were unaware of whether they were in a large or small room.

## Procedure

The experimental procedure was programmed using E-Prime 2.0.

The materials in the experiment were uniformly planned according to the requirements for size, format, and resolution of the picture display; the same display was used for both experimental groups. First, a background with a grey value of 25% was presented on the screen, against which a moral dilemma story and a moral rating scale were displayed in black, italic, 18-point font. 10 moral dilemma stories were presented, either in a large room of 45 square feet or a small room of 16 square feet.

The participants' task was to rate the behavior of the characters in the stories according to their own thoughts and feelings by scoring on a scale ranging from 1 (the character's behavior was moral) to 10 (the character's behavior was immoral).

## Data analysis

**Data screening and processing.** The study retained 60 valid data points that met the requirements of the experiment and analyzed the moral dilemma stories as rated by the two groups of participants in the differently sized rooms. All data were analyzed using SPSS 20.0.

**Statistical analysis of experimental results.** As shown in Table 7, participants placed in the large space environment rated the moral dilemma stories significantly lower ($M = 4.54 \pm 1.18$) than those in the small space environment

**Table 7. Ratings of moral dilemma stories.**

| Experimental condition | Moral dilemma story ratings | t | p |
|---|---|---|---|
| Big space group | 4.54 ± 1.18 | −2.34 | 0.023 |
| Small space group | 5.13 ± 0.71 | | |

($M$ = 5.13 ± 0.71). There was a significant difference between spatial environments in terms of participants' judgments of the moral dilemma story characters' words and actions, $t(58)$ = -2.34, $p$ = 0.023 < 0.05. Participants in the large room group judged the story characters with scores significantly lower than those in the small room group.

## Results and discussion

### The psychological reality of the metaphorical link between the concepts of morality and space on the unconscious level

Individuals activate abstract concepts in their cognition by understanding and processing corresponding concrete concepts. For example, a metaphorical association exists between authority and physical size. People tend to link high authority with bigness and low authority with smallness. This metaphorical mapping originates from sensory experiences of physical dimensions and extends to the abstract concept of authority. Schubert et al. [31] demonstrated that this cognitive mechanism creates a mental association between size and authority, allowing abstract concepts to be more efficiently understood and processed through concrete representations. Based on the above description, Study 1 explored the connection between moral concepts and the size of space. A variant of the Stroop paradigm was employed to verify the association between spatial size perception and abstract moral concepts by presenting various word types in different font sizes. The results revealed that participants exhibited faster response times to moral words displayed in larger fonts and to immoral words presented in smaller fonts. This is consistent with Lakoff and Johnson's (1) Conceptual Metaphor Theory, which posits that 'morality is associated with bigness, whereas immorality is associated with smallness.' Individuals' moral cognitive processing of lexical types originating from the domain of moral concepts activates the size signal in the spatial dimension. Thus, the consistency of concrete concepts (i.e., font sizes) with abstract moral concepts results in a facilitating effect, causing shorter judgment response times and higher accuracy rates. Conversely, the inconsistency of concepts resulted in an inhibitory effect, revealing cognitive conflict in judgment responses, leading to slower response times and lower accuracy rates.

In early physiological development, individuals rely on bodily interactions with the physical and social environment to understand the world. Schwartz, Tesse, and Powell [32] found that an individual's position in the power hierarchy is closely linked to their height and weight, with taller and stronger individuals often perceived as having higher power and social status. Gibbs et al. [33] further clarified that this association arises from inherent physical attributes (e.g., height) rather than learned bodily experiences (e.g., physical training).

Building on this foundation, Schubert [34] explored how humans establish hierarchical relationships and rank within them. Hierarchies require differences in bodily size and strength to form, while size-dimensional cues also shape hierarchical perceptions by activating mental schemas of dominance. This reciprocal relationship indicates that hierarchy judgments are grounded in physical features. In Chinese daily communication, spatial metaphors like *huo da da du* (open-mindedness) and *kang kai da fang* (generosity) use the spatial dimension "big" *(da)* to describe the moral character, whereas terms like *bei bi xiao ren* (despicable villain) and *xiao du ji chang* (small-mindedness) employ "small" *(xiao)* for immoral traits. These linguistic patterns align with Conceptual Metaphor Theory, where moral concepts are systematically mapped onto spatial dimensions, forming the associative principle: *"Big signifies moral, small signifies immoral."* Study 2 utilized an implicit association paradigm to investigate the metaphorical mapping between moral concepts and spatial size. Results showed that participants implicitly associated moral words with larger squares and immoral words with smaller squares. Through rapid reactions, the brain quickly links concepts, thereby deducing the connection between moral concepts and size at the unconscious level. This finding supports the hypothesis that abstract moral concepts and concrete spatial size perceptions are mutually linked through shared mental representations.

When processing moral concepts, individuals automatically activate size-related perceptual schemas derived from spatial source concepts. Specifically, linguistic symbols (e.g., moral/immoral words) trigger size-based metaphorical

mappings, which in turn influence participants' categorization of vocabulary types [35]. Moreover, manipulating concrete-size experiences can prime abstract moral processing and vice versa, suggesting a bidirectional relationship between moral cognition and spatial perception. Building on Lakoff and Johnson's Conceptual Metaphor Theory, Williams, Huang, and Bargh [36] investigated how spatial schemas shape moral cognition. The theory posits that humans use simple, concrete spatial experiences (e.g., verticality, size) to construct intangible concepts like morality. Specifically, there is a systematic mapping between "morality" and "up/bigness" and between "immorality" and "down/smallness." This correspondence arises through shared conceptual projections: positive moral traits are metaphorically linked to expansive spatial dimensions (e.g., "big-hearted"), while negative traits are associated with restrictive dimensions (e.g., "small-minded").

**Impact of the possessive-size spatial metaphor on moral judgment**

Verticality in conceptual metaphors often serves as a foundation for understanding abstract concepts. Spatial-physical information can influence higher-order moral perceptions. Karolina and Bipin [37] experimentally demonstrated that activating vertical concepts impacts moral judgments at both conscious and unconscious levels, in line with Lakoff and Johnson's theory [38] that "up/higher" is metaphorically linked to morality and positivity, while "down/lower" corresponds to immorality and negativity. This suggests that vertical metaphors act as embodied representations through which individuals interact with moral concepts. Building on this, Study 3 hypothesized that spatial size dimensions are conceptually linked to abstract moral judgments. To test this, different spatial sizes were employed to investigate whether individuals activate embodied size perceptions during moral evaluations.

Using an embodied situation test, Study 3 placed participants in either a large or small room to read ambiguous moral dilemmas and evaluate character morality. Participants in the large room exhibited greater leniency and attributed more morality to the characters, whereas those in the small room were harsher in their judgments of immorality. Embodied cognition posits that the environment is an integral part of the cognitive system [39]. The ambiguous nature of the dilemmas was designed to trigger cognitive conflict, prompting participants to rely on environmental cues [28]. Thus, in Study 3, although individuals may appear smaller in a large room, they tend to focus more on the environment's vastness while overlooking cues indicating their own smaller scale. This process activated automatic bodily associations, where spatial size influenced moral evaluations through embodied metaphorical mappings ("morality-bigness," "immorality-smallness"). People draw on pre-existing concepts and information to process complex judgment tasks. Studies 1 and 2 confirmed the psychological reality and bidirectional mappings of metaphors between moral concepts and spatial sizes, demonstrating that people utilize embodied cognitive effects when confronting difficult or ambiguous moral information. Thus, abstract moral evaluations influence spatial perception through metaphorical mappings, forming embodied associations where "big" corresponds to moral and "small" corresponds to immoral.

Finally, it should be noted that while the current investigation has predominantly employed behavioral experimental paradigms, it remains limited in its engagement with cognitive neuroscience frameworks. Future studies could incorporate more embodied interaction tasks combined with relevant physiological measures, providing multidimensional empirical evidence for subsequent research.

## Conclusions

This study illuminates the unconscious psychological relationship underlying size-space metaphors in moral cognition within the framework of embodies cognition, with a specific focus on how spatial environments may shape ethical decision-making (from the variants of Stroop and IAT to the specific physical experience model).

First, the spatial metaphor of the size of moral concepts is psychologically valid at an unconscious level, as individuals characterize abstract moral concepts through concrete size concepts, such as "moral and big, immoral and small." Second, spatial metaphors of bigness and smallness influence individuals' moral judgments. Under the embodiment effect, participants in environments of varying sizes tend to make judgments consistent with the metaphorical associations: "big"

is linked to morality, and "small" is linked to immorality. This study not only provides novel perspectives on moral reasoning, but also offers more empirical insights to advance the theoretical framework of embodied cognition.

## Supporting information

**S1 File. Supporting information files contains S1–S3 dataset.**
(ZIP)

## Acknowledgments

We are grateful for the generous contributions of the research participants and the staff who assisted with data collection during the study.

## Author contributions

**Conceptualization:** Weirui Xiong.

**Data curation:** Weirui Xiong.

**Formal analysis:** Weirui Xiong, Jiaxin Wang, Jiayi Li.

**Methodology:** Weirui Xiong, Jiayi Li.

**Writing – original draft:** Weirui Xiong, Jiayi Li.

**Writing – review & editing:** Jiaxin Wang.

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
