## [Decision Letter · Decision Letter 0]

26 Feb 2025

PONE-D-25-01766Small means immoral? The impact of spatial size metaphor on moral judgmentPLOS ONE

Dear Dr. Xiong,

Thank you for submitting your manuscript to PLOS ONE. After careful consideration, we feel that it has merit but does not fully meet PLOS ONE’s publication criteria as it currently stands. Therefore, we invite you to submit a revised version of the manuscript that addresses the points raised during the review process. In the comments from the two reviewers below, you will see detailed recommendations for how the paper might be improved. Please take into account the recommended revisions from both reviewers.  Please also note that I have acted as a reviewer for this manuscript, and you will find my comments as Reviewer 1. Please submit your revised manuscript by Apr 12 2025 11:59PM. If you will need more time than this to complete your revisions, please reply to this message or contact the journal office at plosone@plos.org . Please include the following items when submitting your revised manuscript:

We look forward to receiving your revised manuscript.

Kind regards,

Kevin Schilbrack

Academic Editor

PLOS ONE

**Comments to the Author**

1. Is the manuscript technically sound, and do the data support the conclusions?

Reviewer #1: Yes

Reviewer #2: Yes

2. Have the authors made all data underlying the findings in their manuscript fully available?

Reviewer #1: Yes

Reviewer #2: Yes

3. Is the manuscript presented in an intelligible fashion and written in standard English? (PLOS ONE does not copyedit accepted manuscripts, so the language in submitted articles must be clear, correct, and unambiguous.)

Reviewer #1: No

Reviewer #2: No

4. Review Comments to the Author

Reviewer #1: This paper describes three psychological experiments that seek to show a connection between moral concepts and physical space. One experiment examined a relationship between a large and small fonts used for moral and immoral words, another examined a relationship between moral and immoral words and larger and smaller squares, and the third examined how participants responded to moral dilemmas when in a bigger or smaller room. I think that the conceptual metaphor that “big is good” is already well-established, and so the paper would be stronger if it began with some explanation of what the paper contributes that is new. Nevertheless, the paper discusses an interesting and appropriate topic for PLOS One, and it could be published once revised. However, conceptual problems and grammar problems mean that the revisions would be many.

**Grammatical problems**

This paper has a significant number of grammar problems. I saw words mis-spelled, verb agreement problems, and sentences without punctuation. The paper speaks of “bodily size” instead of body size (p. 17). In several cases, the sentences are ungrammatical to the point that the paper could not be published unless they were corrected, but the problems do not obscure the meaning intended. Here are some examples of this.

1. The following sentence is missing its articles: “Source concept is direct experience in daily lives” (p. 2). This should be “A source concept is a direct experience in a person’s daily life” or, better, since a concept is not an experience, “A source concept is found in a direct experience in a person’s daily life.”

2. In some cases, the verbs are not idiomatic. For example, the paper says that Conceptual Metaphor Theory “believes” that understanding is such-and-such (p. 2). But a theory is not a person and so it cannot believe things. It would be better to say: According to Conceptual Metaphor Theory, understanding is such-and-such.

3. The paper says that “Merleau-Ponty proposed that the importance of the body and senses arising within organizational behavior” (p. 3). However, this is not a complete sentence and so I am not sure what the authors are trying to say. I think that the authors meant to write that Merleau-Ponty wrote about the importance of the body and senses arising within organizational behavior.

4. One experiment describes concepts “of two different sizes (8 cm x 8cm and 4 cm x cm)” (p. 10). I think that the two different sizes described here are actually squares, not concepts, and I imagine that the squares are made of paper, but here again the paper was hard to follow.

However, the English is often ungrammatical to the point that the paper’s argument is obscured. Here are some examples of this.

1. The authors write that “the words in a bigger font gave a more positive meaning” (p. 3) but what they mean to say is the importantly different claim that the bigger font of the words gave a more positive meaning.

2. The paper says that certain scholars were “founding that participants would increase the perceived size of object” (p. 3). Setting aside the problem that the word “object” here is missing its article, the word “founding” does not make sense in this context. The authors probably mean something like “speculating.” However, perhaps they mean the very different idea that the cited paper “found” that participants did this, in the sense that the scholars discovered or showed it.

3. The paper says this: “when the participant was deprived of the freedom to drink water, the subject would perceive a bigger cup size” (p. 3). But what the authors mean that the scholars hypothesized that when the participant was deprived of the freedom to drink water, the subject would perceive a bigger cup size. By failing to include the phrase “They hypothesized that,” the paper turns a hypothesis into a fact.

4. I could not figure out what is meant by this sentence: “A priori power is consistent with the above” (p. 9). Of course, “a priori” is a Latin phrase that is important in philosophy, but I am not sure what is meant by “a priori power.” The paper is not making a philosophical argument, and it had never introduced the idea of “a priori power” or even the idea of power. Later, the paper says this: “A priori power analysis is consistent with the above” (p. 14). This makes me suspect that the word “analysis” was dropped from the first sentence, but I don’t know what “a priori power analysis” or even what “power analysis” is.

Most of these examples were taken from the first pages of the paper, and then I stopped marking grammar problems. I think that the authors need to have the paper read sentence by sentence by a native English speaker to make sure that the paper is representing their ideas accurately.

**Conceptual problems**

None of these conceptual problems is severe.

1. In the abstract, the authors claim that the paper shows that there is an unconscious reality between moral concepts and size (pp. 1, 4). I think that they do not mean (or, at least, they should not mean) that they have discovered a reality between moral concepts and size. I think that they mean to claim that there is a *relationship * between moral concepts and size.

2. The paper says “In english expressions, there are still some words (such as large, wide, broad) to describe moral behavior” (p. 2). I was confused why the paper says that these expressions “still” exist, as if such expressions are old-fashioned or being eliminated. And though the claim is likely to be correct, I cannot think of any expressions for moral behavior with these terms, and so I wished that the paper had said what they were.

3. The describe a contrast between “self-purification” and “self-cleansing” (p. 3). These seem identical and I think that the problem is that the paper was written incorrectly.

Reviewer #2: The studies described in this article—particularly Study 3—warrant dissemination as they extend the knowledge we have about how humans make moral judgements, and how such judgments are embodied. However, the article is not currently written to make that contribution clear.

The first half of the literature review lacks organization and precision. It moves from similar studies designed to show the connection between size and morality, to general remarks about the place of embodiment (e.g. Merleau-Ponty, whose connection to embodied cognition needs to be nuanced since he predates it), to an assortment of embodied cognition studies. Within this section there is an elision between conceptual metaphor theory and embodied cognition—not all advocates of CMT advocate all theories of embodied cognition, and there are different theories of embodied cognition. Right now “embodied cognition” is defined without citation within the discussion on Merleau-Ponty. I don’t think a paper like this needs to necessarily wade into these theoretical waters, but if it does the section should be clearer and more developed.

More critically, the discussion of how the current studies build on this literature is underdeveloped. For example, in discussing Study 1 vis-à-vis Lu, Guo and Jiang (2017) the article states that “Nonetheless, the existence of an unconscious mental association with moral conceptual size-space metaphors has yet to be empirically substantiated, nor has their metaphorical consistency been rigorously examined within the context of complex, embodied moral judgment tasks. Consequently, study 1 verifies the psychological reality of the existence of metaphors between moral concepts and big and small spaces on an unconscious level…” but the study seems to be replicating Lu, Guo and Jiang. Multiple times in the article the difference is cached out in terms of unconscious vs conscious connections (see for example the end of the discussion of study 3) but it’s not clear how or why Studies 1 and 2 deal with the unconscious whereas studies like Lu, Guo, and Jiang (2017) deal with the conscious.

Study 3 is the most original, yet it is given the least explanation (this may be an organization problem, as I wonder if lines like the one I quoted in the paragraph above weren’t supposed to be specifically about Study 3, insofar as it specifically is dealing with spaces). Whereas the Stroop Test and the IAT have the subject engaging size depicted on screens, Study 3 examines the size of the space the subject is in. This difference seems particularly relevant given the authors’ interest in embodied cognition, as one could argue that the results are more suggestive of the brain’s hijacking of sensorimotor systems for moral reasoning (vs. merely “associating” size and morality). (And if this is what the authors’ mean by conscious vs. unconscious that needs to be better explained, and clarified that it is Study 3 that specifically offers this contribution).

There is another important consideration—whereas in Studies 1 and 2 it is clear that bigness and morality are being linked, it is not so clear once we are putting a subject in a room. Critically, a person in a big room may feel smaller; the screen that they are reading the moral dilemma off will also appear relatively smaller. So the design of Study 3 introduces some complexity that the authors need to address. In other words I don’t think it’s a straight of a line from seeing a big box on a screen to being in a big box.

With that being said, if this article were rewritten to address these issues I think the main contribution—of moving beyond Stroop(-like) Tests and IATs in establishing the principles of CMT and certain theories of embodied cognition to more complicated models of how bodies move through space—is an important one and worth publishing.

6. PLOS authors have the option to publish the peer review history of their article (what does this mean? ). If published, this will include your full peer review and any attached files.

**Do you want your identity to be public for this peer review?** For information about this choice, including consent withdrawal, please see our Privacy Policy .

Reviewer #1: No

Reviewer #2: No

---

## [Author Response · Author response to Decision Letter 1]

22 Mar 2025

Dear editor:

We appreciate very much your insightful comments and critiques. These comments are very helpful for us to revise this paper. We have carried out a comprehensive revision on the manuscript in accordance to your comments. The revisions were addressed point by point below. For the convenience of review, I marked all the changes in red fonts in the revised manuscript.

Response: Thank you for your valuable comments. We have already made adjustments according to the template.

Response: Thank you for your valuable comments. We have already added the full name of ethics committee and the information of informed written consent from participants in Materials and Methods of Study 1. The changes as requested are as follows:

All studies were approved by the Ethics Committee of Chongqing Normal University. All informed consent forms were signed before participants completed the questionnaire for evaluation and began the studies. All experimental procedures adhered to the ethical guidelines outlined in the Declaration of Helsinki.

Response: Thank you for your valuable comments. This article does not have supporting information.

Review Comments to the Author

Reviewer #1: This paper describes three psychological experiments that seek to show a connection between moral concepts and physical space. One experiment examined a relationship between a large and small fonts used for moral and immoral words, another examined a relationship between moral and immoral words and larger and smaller squares, and the third examined how participants responded to moral dilemmas when in a bigger or smaller room. I think that the conceptual metaphor that “big is good” is already well-established, and so the paper would be stronger if it began with some explanation of what the paper contributes that is new. Nevertheless, the paper discusses an interesting and appropriate topic for PLOS One, and it could be published once revised. However, conceptual problems and grammar problems mean that the revisions would be many.

Response: Thank you for your valuable comments. We have revised the manuscript, providing more detailed explanations to address conceptual issues and further polishing the grammar with additional edits and refinements.

Grammatical problems

This paper has a significant number of grammar problems. I saw words mis-spelled, verb agreement problems, and sentences without punctuation. The paper speaks of “bodily size” instead of body size (p. 17). In several cases, the sentences are ungrammatical to the point that the paper could not be published unless they were corrected, but the problems do not obscure the meaning intended. Here are some examples of this.

Response: Thank you for your valuable comments. We have revised the vocabulary, grammar, and punctuation of the paper, and restructured complex sentences to ensure the key arguments remain unobscured.

1. The following sentence is missing its articles: “Source concept is direct experience in daily lives” (p. 2). This should be “A source concept is a direct experience in a person’s daily life” or, better, since a concept is not an experience, “A source concept is found in a direct experience in a person’s daily life.”

Response: Thank you for your valuable comments. The changes as requested are as follows:

A source concept originates from a direct experience in a person's daily life, whereas a target concept is an abstract and vague idea, entity, or emotion [1].

2. In some cases, the verbs are not idiomatic. For example, the paper says that Conceptual Metaphor Theory “believes” that understanding is such-and-such (p. 2). But a theory is not a person and so it cannot believe things. It would be better to say: According to Conceptual Metaphor Theory, understanding is such-and-such.

Response: Thank you for bringing it to our attention. The error has been corrected. This sentence has been modified:

According to CMT, understanding target concepts is facilitated by drawing on well-known and clearly represented prior experiences (source concepts).

3. The paper says that “Merleau-Ponty proposed that the importance of the body and senses arising within organizational behavior” (p. 3). However, this is not a complete sentence and so I am not sure what the authors are trying to say. I think that the authors meant to write that Merleau-Ponty wrote about the importance of the body and senses arising within organizational behavior.

Response: Thank you very much for pointing it out. The changes as requested are as follows:

With the development of experiential philosophy and phenomenology, Merleau-Ponty [4] emphasized that the importance of the body and senses arises within organizational behavior.

4. One experiment describes concepts “of two different sizes (8 cm x 8cm and 4 cm x cm)” (p. 10). I think that the two different sizes described here are actually squares, not concepts, and I imagine that the squares are made of paper, but here again the paper was hard to follow.

Response: Thank you for your advice. This has been added to the article and your help in improving it is much appreciated. The changes as requested are as follows:

The target concepts included 10 moral and 10 immoral words, while the source concepts consisted of two squares of different sizes (8 cm x 8 cm and 4 cm x 4 cm). All materials were presented on a computer screen.

However, the English is often ungrammatical to the point that the paper’s argument is obscured. Here are some examples of this.

1. The authors write that “the words in a bigger font gave a more positive meaning” (p. 3) but what they mean to say is the importantly different claim that the bigger font of the words gave a more positive meaning.

Response: Thank you for your valuable comments. This sentence has been modified:

The larger font of the words conveyed a more positive meaning, illustrating the metaphorical association between 'big and positive' and 'small and negative".

2. The paper says that certain scholars were “founding that participants would increase the perceived size of object” (p. 3). Setting aside the problem that the word “object” here is missing its article, the word “founding” does not make sense in this context. The authors probably mean something like “speculating.” However, perhaps they mean the very different idea that the cited paper “found” that participants did this, in the sense that the scholars discovered or showed it.

Response: We would like to apologize for our oversight. Thank you for bringing it to our attention. The error has been corrected. This sentence has been modified:

Martijn, Henk, and Ruud [3] examined the desire to achieve the goal and found that participants would increase the perceived size of an object, which would help achieve the goal.

3. The paper says this: “when the participant was deprived of the freedom to drink water, the subject would perceive a bigger cup size” (p. 3). But what the authors mean that the scholars hypothesized that when the participant was deprived of the freedom to drink water, the subject would perceive a bigger cup size. By failing to include the phrase “They hypothesized that,” the paper turns a hypothesis into a fact.

Response: Thank you for bringing it to our attention. We have already added “the researchers hypothesized that”. This sentence has been modified:

For instance, the researchers hypothesized that when a participant was deprived of the freedom to drink water, they would perceive a cup as larger.

4. I could not figure out what is meant by this sentence: “A priori power is consistent with the above” (p. 9). Of course, “a priori” is a Latin phrase that is important in philosophy, but I am not sure what is meant by “a priori power.” The paper is not making a philosophical argument, and it had never introduced the idea of “a priori power” or even the idea of power. Later, the paper says this: “A priori power analysis is consistent with the above” (p. 14). This makes me suspect that the word “analysis” was dropped from the first sentence, but I don’t know what “a priori power analysis” or even what “power analysis” is.

Response: Thank you for your advice. This has been added to the article and your help in improving it is much appreciated. The changes as requested are as follows:

Study 1�According to calculation using G*power 3.1 [25], for the two-way repeated measures analysis of variance applicable to this study, with a significance level of α = 0.05, a large effect size (f = 0.4), and a predicted statistical power (1-β) of 0.8, the required total number of research subjects was 22.

Study 2�According to calculation using G*power 3.1 [25], for the Paired samples t-tests applicable to this study, with a significance level of α = 0.05, a large effect size (f = 0.8), and a predicted statistical power (1-β) of 0.8, the required total number of research subjects was 15.

Study 3�According to calculation using G*power 3.1 [25], for the independent samples t-test applicable to this study, with a significance level of α = 0.05, a large effect size (d= 0.8), and a predicted statistical power (1-β) of 0.8, the required total number of research subjects was 52.

Conceptual problems

None of these conceptual problems is severe.

1. In the abstract, the authors claim that the paper shows that there is an unconscious reality between moral concepts and size (pp. 1, 4). I think that they do not mean (or, at least, they should not mean) that they have discovered a reality between moral concepts and size. I think that they mean to claim that there is a relationship between moral concepts and size.

Response: Thank you for your valuable comments. We have changed word to relationship. This sentence has been modified:

Collectively, these three studies demonstrate that the unconscious psychological relationship between moral concepts and the spatial dimension of size influences individuals’ abstract moral judgments under embodied cognition.

2. The paper says “In english expressions, there are still some words (such as large, wide, broad) to describe moral behavior” (p. 2). I was confused why the paper says that these expressions “still” exist, as if such expressions are old-fashioned or being eliminated. And though the claim is likely to be correct, I cannot think of any expressions for moral behavior with these terms, and so I wished that the paper had said what they were.

Response: Thank you for your valuable comments. We have removed 'still' from the sentence. This sentence has been modified:

Similarly, in English, certain words such as large, wide, and broad are used to describe moral behavior, while words such as small, narrow, and thin are often associated with immoral behavior. People comprehend abstract concepts not only through concrete notions of size but also by recognizing the correlation between size and emotional valence.

3. The describe a contrast between “self-purification” and “self-cleansing” (p. 3). These seem identical and I think that the problem is that the paper was written incorrectly.

Response: We would like to apologize for our oversight. Thank you for bringing it to our attention. We have changed "self-purification" to "self-dirtiness". This sentence has been modified:

Within the framework of the purification metaphor, Ding and Wang [8] observed that participants primed with self-dirtiness exhibited heightened reactivity to negative moral emotion words compared to those primed with self-cleansing.

Reviewer #2: The studies described in this article—particularly Study 3—warrant dissemination as they extend the knowledge we have about how humans make moral judgements, and how such judgments are embodied. However, the article is not currently written to make that contribution clear.

Response: Thank you very much for pointing it out. Interpretation of the introduction has been added as described below:

These studies aim to expand knowledge on how humans make moral judgments and how these judgments are embodied. The anticipated contributions lie in bridging gaps between unconscious cognitive processes and embodied metaphor theory, potentially informing interdisciplinary discussions in moral psychology, linguistics, and environmental design.

The first half of the literature review lacks organization and precision. It moves from similar studies designed to show the connection between size and morality, to general remarks about the place of embodiment (e.g. Merleau-Ponty, whose connection to embodied cognition needs to be nuanced since he predates it), to an assortment of embodied cognition studies. Within this section there is an elision between conceptual metaphor theory and embodied cognition—not all advocates of CMT advocate all theories of embodied cognition, and there are different theories of embodied cognition. Right now “embodied cognition” is defined without citation within the discussion on Merleau-Ponty. I don’t think a paper like this needs to necessarily wade into these theoretical waters, but if it does the section should be clearer and more developed.

Response: Thank you for your advice. This has been added to the article and your help in improving it is much appreciated. We made that all theories discussed in the article were presented within the framework of Embodied Cognition and add some sentences. Interpretation of the introduction has been added as described below:

Within the framework of embodied cognition, Lakoff and Johnson [1] proposed the Conceptual Metaphor Theory (CMT), which suggests that the conceptual systems underlying human thought, cognition, and behavior are structured by metaphors.

Furthermore, conceptual representation, as a fundamental and primary cognitive ability of humans, is inherently linked to the individual's perceptual-motor system and cannot exist independently of it [5].

Embodied cognition theory holds that cognitive activities are not only inseparable from the brain but are also closely linked to the interaction between the body and the environment.

More critically, the discussion of how the current studies build on this literature is underdeveloped. For example, in discussing Study 1 vis-à-vis Lu, Guo and Jiang (2017) the article states that “Nonetheless, the existence of an unconscious mental association with moral conceptual size-space metaphors has yet to be empirically substantiated, nor has their metaphorical consistency been rigorously examined within the context of complex, embodied moral judgment tasks. Consequently, study 1 verifies the psychological reality of the existence of metaphors between moral concepts and big and small spaces on an unconscious level…” but the study seems to be replicating Lu, Guo and Jiang. Multiple times in the article the difference is cached out in terms of unconscious vs conscious connections (see for example the end of the discussion of study 3) but it’s not clear how or why Studies 1 and 2 deal with the unconscious whereas studies like Lu, Guo, and Jiang (2017) deal with the conscious.

Response: We have made the revisions according to your suggestions. We sincerely appreciate your correction, which has enabled our manuscript to be

---

## [Editor Report · Decision Letter 1]

6 May 2025

Small means immoral? The impact of spatial size metaphor on moral judgment

PONE-D-25-01766R1

Dear Dr. Xiong,

We’re pleased to inform you that your manuscript has been judged scientifically suitable for publication and will be formally accepted for publication once it meets all outstanding technical requirements.

Kind regards,

Kevin Schilbrack

Academic Editor

PLOS ONE
---

## [Editor Report · Acceptance letter]

PONE-D-25-01766R1

PLOS ONE

Dear Dr. Xiong,

I'm pleased to inform you that your manuscript has been deemed suitable for publication in PLOS ONE. Congratulations! Your manuscript is now being handed over to our production team.

Kind regards,

on behalf of

Dr. Kevin Schilbrack

Academic Editor

PLOS ONE